# Reproducibility report - DECAF: Generating Fair Synthetic Data Using Causally-Aware Generative Networks

## Reproducibility Summary

**Scope of Reproducibility**

In the DECAF paper [2], the authors introduce a causal GAN-based model for generating fair synthetic data. Additionally, the paper describes a flexible causal approach for modifying this model such that it can generate fair data. Furthermore, van Breugel et al. guarantee that downstream models trained on the generated synthetic data, can generate fair predictions on both synthetic and real data. We aim to reproduce these claims.

**Methodology**

We've started off using the original codebase provided by the authors of the DECAF paper and we've started trying to reproduce the results mentioned in the original paper. Our main focus was in trying to use the GAN-based DECAF model for generating fair synthetic data. This had to be done before we could get the other results that contained some effort of debiasing. We've tried this out on the adult dataset and the credit approval dataset both of which were mentioned by the authors of the original paper. Additionally we have tried to reproduce the causal graph discovered of the credit approval dataset using fast greedy equivalence search.

**Results**

The DECAF model proposed by the authors was trained on the adult dataset and the credit approval dataset. We used the original hyperparameters proposed by the paper.
The architecture proposed by the paper included causal graphs - this was not included in the GitHub repository.
The paper specified architectures to compare to DECAF, however, it did not include code which allowed us to do so. Instead, a GAN architecture was used to generate new, synthetic data, which allowed us to compare DECAF to another method.

**What was easy**

The authors describe in a clear manner how to implement the removal of edges in the Adult datasetś DAG in order to satisfy the fairness conditions. It was therefore easy to implement this for each of the fairness definitions in the DECAF paper [2]. Furthermore, the paper's instruction on how to perform the evaluation are also clear and easy to implement.

**What was difficult**

Most of the difficulty lies in trying to get the GAN based framework to get proper results. This also had to do with the fact that we did not know that we had to provide the model with a DAG seed and a bias dict. So efforts were made into manually encoding the connections of the graph based on the graph provided in the paper.

**Communication with original authors**

No contact was sought with the authors of the paper. Contact between peers working on the same project has been most fruitful in the sense that we we're able to get knowledge into how they dealt with the issues that we were both facing.

33rd Conference on Neural Information Processing Systems (NeurIPS 2020), Vancouver, Canada.

# 1 Introduction

In recent years, the focus on fairness in artificial intelligence has gained increased attention. Seeing as how more advanced algorithms are being used for tasks of ever-increasing importance, the ensurance of fair, balanced algorithms is a must if we are to progress to a more egalitarian society.

The problem with modern machine learning architectures is that they are able to capture highly complex patterns in data, which generally results in high performance and low transparency. Logically, a complex model has a large number of parameters which, to the human mind, are very hard or even impossible to decipher. Therefore, if a model is tasked with generating predictions, it is non-trivial to justify a singular prediction, resulting in diminished insight in a model's fairness.

In their paper, van Breugel et al. propose a novel approach to machine learning while also guaranteeing a reduction in bias in any tabular data. Given a dataset, they claim to have devised a method that can generate new, synthetic data that has reduced influence from a protected attribute (race, gender, age, etc.) and can still be considered relatively faithful to the original dataset. This method offers different degrees of privacy, that is, how much of the influence of the protected attribute we remove, with a limited decrease in terms of likeness to the original dataset.

The provided GitHub repository provided a much-needed basis for replicating the paper. The main DECAF architecture was specified, implemented, and the repository contained instructions on how to run the code. When running the specified test commands, the program gave off errors that did not have straight-forward fixes. It was also not immediately possible to generalize the program; the causal graphs that are essential to the proposed architecture were not generated by the code. (There was a hard-coded example and a mention of the causal graph discovery algorithms used.)

# 2 Scope of reproducibility

The original paper by van Breugel et al.[2] introduces DECAF a GAN-based model for generating fair synthetic data. The data generating process is embedded as a structural causal model in the input layers of the model, allowing each variable to be reconstructed conditioned on its causal parents. Additionally, the designed framework allows for inference time debiasing meaning that biased edges can be removed in order to satisfy user-defined fairness requirements. Thus, paper's main contributions can be formalized in three claims:

1. DECAF, a causal GAN-based model for generating synthetic data.
2. a flexible causal approach for modifying this model such that it can generate fair data.
3. the guarantee that downstream models trained on the synthetic data, will also generate fair predictions in other settings.

Our aim was to reproduce the evaluation performed in the paper in order to demonstrate these claims. To achieve this goal, we attempt to reproduce the evaluation condition using the provided code. Additionally, we reproduce the FactGAN model in order to compare the results of the original paper's evaluation as well as our own DECAF evaluation with the generated benchmark results.

# 3 Methodology

This section describes the methodology of trying to follow the steps taken for generating synthetic data and deliberately debiasing it in order to let any downstream classifier ensure fairness of the model if trained on the debiased synthetic data.

## 3.1 Model descriptions

### 3.1.1 DECAF

The main proposed contribution in the DECAF paper (claim 1) is a causal GAN-based model for generating synthetic data. GAN (Generative Adversarial Networks) are based on two simultaneously trained deep network models: a generator and a discriminator. These two networks compete with each other to reach a higher accuracy in their predictions [1]. GAN models focus on generating data from scratch. This is usually done for images but it can also be

applied to other fields. In this case, GANs are used for tabular data generation. Furthermore, the GAN models used in the paper's experiment are described as causal-informed GANs. This is an important aspect of the paper as it aims to generate fair synthetic data by leveraging causal structures such as DAGs (Directed acyclic graph). Indeed the training phase of the DECAF model consist of a causally informed GAN to learn the causal conditions. While, the inference phase consist of the generator creating fair data based on one of 3 corollaries for debiasing. These 3 corollaries describe how to remove the bias in a DAG structure in order to satisfy one of the 3 corresponding definition of fairness [2]:

- Fairness Through Unawareness (FTU): A predictor f : $X \rightarrow \hat{Y}$ is fair iff protected attributes A are not explicitly used by f to predict $\hat{Y}$.

- Demographic Parity (DP): A predictor $\hat{Y}$ is fair iff $A \perp\!\!\!\perp \hat{Y}$, i.e. $\forall a, a' : P(\hat{Y}|A = a) = P(\hat{Y}|A = a')$.

- Conditional Fairness (CF): A predictor $\hat{Y}$ is fair iff $A \perp\!\!\!\perp \hat{Y}|R$, i.e. $\forall a, a' : P(\hat{Y}|R = r, A = a) = P(\hat{Y}|R = r, A = a')$.

Based on these variations, multiple versions of DECAF are evaluated in the paper: DECAF-ND, DECAF-FTU, DECAF-DP and DECAF-CF. DECAF-ND stands for DECAF with no debiasing.

In order to showcase claim 3 which guarantees that downstream classifiers trained on the generated synthetic data will also provide fair predictions in other settings, a MLP (Multilayer perceptron) was trained using the scikit-learn package [2].

### 3.1.2 CTGAN

In order to create some contrast for DECAF, another GAN-based model was considered. *CTGAN*[1], a python library by sdv, is a conditional GAN to have some comparison to the DECAF architecture. It is designed to generate new data conditioned on a tabular dataset, meaning we can exclude data to generate a attribute without it being conditioned on a node further down the causal graph. (And we can also exclude nodes that are not to be used when including privacy metrics.)

### 3.1.3 FAIRGAN

Eventually code was obtained which contained code for reproducing fairGAN benchmark results. FairGAN is one of the main models that tries to incorporate fairness into the data generation process therefore it is an interesting result to have for comparison with other results obtained in the process of reproducing this paper.
The codebase was mainly adapted towards computing fairGAN benchmarks on the credit approval dataset. It is done by introducing synthetic bias into the original credit approval dataset through modifying the original dataset such that a particular ethnicity is always rejected a loan.
The GAN is trained across 50 epochs using a standard learning rate scheduler and the model is implemented in tensorflow. The score shown in the results section are obtained by averaging over 10 runs computing the mean and standard deviation across all runs. Also, bias is induced with a bias value $\beta = 1.0$. This value regulates how strong the bias is that is synthetically induced in an unbiased dataset. The codebase also contains code for generating synthetic data for various different bias values. These can be obtained through running the fairGAN code.

## 3.2 Datasets

Experiments were conducted on the adult dataset [2] and the credit approval dataset [3]. The adult dataset is the main dataset on which the experiments were conducted. A graph seed of this dataset has to be provided to the framework which specifies the causal relations that the variables have with one another. Also the framework needs the connections that need to be removed in order for the different notions of fairness to be satisfied. This information is however specifically provided in the paper however the authors state this comments left later on questions that reviewers of the paper had.

The adult dataset contains 48842 entries across 14 different labels which are both continuous and discrete. It contains census data of individuals whose income exceeds $50k/yr or not. It has been published in 1996 and contains entries such as **race, age, sex, occupation and education**.

The credit approval dataset contains credit card applications. It contains 690 entries and 15 attributes of a mix between continuous and discrete variables. All variables as well as the entry names have been changed to meaningless symbols

---

[1]https://sdv.dev/SDV/user$_g$uides/single$_t$able/ctgan.html
[2]https://archive.ics.uci.edu/ml/datasets/Adult
[3]https://archive.ics.uci.edu/ml/datasets/credit+approval

to protect the confidentiality of the data. It also contains a few missing values. Also the adult dataset was split into an X and a y, where x contained the tabular entries and y corresponded to label of the individual that either had an income larger that 50k or lower than 50k.

Pre-processing of the dataset consisted of replacing all discrete entries by numerical values where every possible discrete entry is assigned a numerical value in the range of all possible discrete entries for that variable. The same was done for the credit approval dataset. For the credit approval dataset the labels were translated back into the original (non-anonymised) labels through making use of an online source [4].

All data was used to train the xgb classifier. No train/test split was done since no validation was executed on the model.

### 3.3 Hyperparameters

The DECAF model used throughout our replication of the paper's experiments is composed of a hidden dimension of 200 neurons. It utilizes a ReLU activation function, the Adam algorithm as optimizer and a learning rate of 0.001 [2].

The MLP used as downstream classifier, is composed of a hidden layer with 100 neurons and is trained using a ReLU activation function. The output layer is trained using a softmax activation, a binary cross entropy loss function, the Adam algorithm as optimizer and a learning rate of 0.001 [2].

The authors of [2] state in a response to reviews left about their paper that users of their framework should provide the model with a DAG-seed and a bias-dict [5]. The seed provided should contain the conditional dependence of the variables in the dataset. The bias-dict should specify the dependencies that need to be removed in order for a notion of fairness to be satisfied. The paper does not explicitly mention that these variables should be provided to the model.

Throughout the experiments conducted the data is normalized using a min-max scaler provided by the sklearn library. The DECAF framework is provided an input dimension which corresponds to the number of variables in the dataset excluding the label. Use_mask is set to true, grad_dag_loss is set to False, lambda privacy is set to 0, Weight decay is set to 0.01 l1_g is set to 0 and p1_gen is set to -1. A batch size of 100 is used.

The rest of the hyperparameters are left unchanged and standard according to the default tuning. By default it uses a hidden layer size of 200, a learning rate of 0.001 and a batch size of 32. It uses a standard *b1* and *b2* of 0.5 and 0.999 respectively.

Lambda_gp equals 10, *Eps* is $1e-8$, Alpha and Rho are both 1, l1_w is 1.

### 3.4 Experimental setup and code

In order to replicate the experiments the author's original code was utilized. All experiments are conducted in a Python virtual environment with Python 3.6. Instructions to replicate the evaluation environment can be found in the read me file in our code repository. All experiments are provided on a single Jupyter notebook file that can be found in the repository. For all the experiments that require a deep learning framework, the PyTorch library was utilized.

Two main metrics are of interest in order to evaluate the experiments: the quality and the fairness of the generated synthetic dataset. In order to assess the data quality, the precision and recall of the prediction on the synthetic dataset is computed against the predictions of the original dataset. Additionally, the AUROC (area under the receiver operating characteristic) of predicting the target variable using a downstream classifier trained on synthetic data is also used as performance metric. The AUROC tells us the model's ability to discriminate between positive and negative cases. The fairness is computed using the FTU (Fairness Through Unawareness) algorithm and the DP (Demographic Parity) algorithm. The FTU score is computed by calculating the difference between the predictions of a downstream classifier for setting A to 1 and 0, respectively, such that $|P_{A=0}(\hat{Y}|X) - P_{A=1}(\hat{Y}|X)|$, while keeping all other features the same. This difference measures the direct influence of A on the prediction. The DP score is measured in terms of the Total Variation: the difference between the predictions of a downstream classifier in terms of positive to negative ratio between the different classes of protected variable A [2].

Using the same downstream classifier, this paper ultimately compares different datasets in terms of accuracy and fairness. That is, the original dataset and synthetic data generated by different GAN models. The original codebase provided along with the paper did not contain any model other than DECAF, meaning the comparison had to be constructed from scratch. See sections 3.1.2 and 3.1.3 for more details.

The CTGAN model, desribed in section 3.1.2 uses the same environment as the DECAF model.

---

[4] https://bit.ly/3G8FCoi
[5] https://openreview.net/forum?id=XN1M27T6uux

However, a separate environment had to be created for the evaluations of the FairGAN model as it relies on Tensorflow 1, and the DECAF code relies on Pytorch-lightning 1.4.*. Both libraries use different versions of tensorboard which leads to a dependecy conflict issue. Clear instruction are provided in order to replicate the FairGAN environment, in the same manner as the instructions to create the DECAF environment.

Our code and instructions can be found in the following repository: `https://anonymous.4open.science/r/fact-ai-CODD/README.md`

### 3.5 Computational requirements

All the conducted experiments were conducted locally. Optimizing the training of our models using a GPU was not necessary in our case as a CPU was sufficient.

## 4 Results

In this section, the results of the conducted replication and evaluations are presented.

### 4.1 Results reproducing original paper

#### 4.1.1 Result: bias removal experiment

In order to replicate the main experiment of the paper and verify claim 1, the implementation of the DECAF model is trained using the adult dataset [6] and the hyperparameters described in the original experiments. Table 2 shows the results from our conducted experiment, and table 1 showcases the ones from the original paper. The table also contains some results on benchmarks for comparison. In comparison to the original paper the results are quite different. The precision of our reproducal is around half of what has been obtained in the original paper. Recall can be witnessed is also much lower. Fairness through awarenes curiously is around the same value.

| Method | Precision | Recall | AUROC | FTU | DP |
|---|---|---|---|---|---|
| Original data D | $0.920 \pm 0.006$ | $0.936 \pm 0.008$ | $0.807 \pm 0.004$ | $0.116 \pm 0.028$ | $0.180 \pm 0.010$ |
| GAN | $0.607 \pm 0.006$ | $0.439 \pm 0.037$ | $0.567 \pm 0.132$ | $0.023 \pm 0.010$ | $0.089 \pm 0.008$ |
| WGAN-GP | $0.683 \pm 0.015$ | $0.914 \pm 0.005$ | $0.798 \pm 0.009$ | $0.120 \pm 0.014$ | $0.189 \pm 0.024$ |
| FairGAN | $0.681 \pm 0.023$ | $0.814 \pm 0.079$ | $0.766 \pm 0.029$ | $0.009 \pm 0.002$ | $0.097 \pm 0.018$ |
| DECAF-ND | $0.780 \pm 0.023$ | $0.920 \pm 0.045$ | $0.781 \pm 0.007$ | $0.152 \pm 0.013$ | $0.198 \pm 0.013$ |
| DECAF-FTU | $0.763 \pm 0.033$ | $0.925 \pm 0.040$ | $0.765 \pm 0.010$ | $0.004 \pm 0.004$ | $0.054 \pm 0.005$ |
| DECAF-CF | $0.743 \pm 0.022$ | $0.875 \pm 0.038$ | $0.769 \pm 0.004$ | $0.003 \pm 0.006$ | $0.039 \pm 0.011$ |
| DECAF-DP | $0.781 \pm 0.018$ | $0.881 \pm 0.050$ | $0.672 \pm 0.014$ | $0.001 \pm 0.002$ | $0.001 \pm 0.001$ |

Table 1: Bias removal experiment on the Adult dataset (Original results)

| Method | Precision | Recall | AUROC | FTU | DP |
|---|---|---|---|---|---|
| FairGAN | $0.328 \pm 0.218$ | $0.308 \pm 0.243$ | $0.856 \pm 0.05$ | $0.009 \pm 0.051$ | $0.324 \pm 0.313$ |
| DECAF-FTU | 0.938 | 0.794 | 0.722 | 0.285 | 0.336 |
| DECAF-CF | 0.831 | 0.999 | 0.500 | 0.004 | 0.0001 |
| DECAF-DP | 0.962 | 0.968 | 0.573 | 0.046 | 0.055 |
| CTGAN-FTU | 0.752 | 0.826 | 0.502 | 0.060 | 0.063 |
| CTGAN-CF | 0.750 | 0.929 | 0.499 | 0.063 | 0.061 |
| CTGAN-DP | 0.751 | 0.873 | 0.500 | 0.092 | 0.093 |

Table 2: Bias removal experiment on the Adult dataset (Reproduced evaluation)

Firstly, when comparing the two table you can immediately notice the considerable contrast between our DECAF results and the ones from the original paper. It is clear that the DECAF evaluations are not reproducible in the same exact manner as the authors performed them using only the code that was provided. We therefore tried to implement our own DECAF model using CTGAN. (see section 4.2)

---

[6]https://archive.ics.uci.edu/ml/datasets/Adult

### 4.1.2 Result: causal graph discovery

In the paper the authors mention using fast greedy equivalence search for discovering a causal graph based on the credit approval dataset. So far we have not been able to generate the identical graph displayed in the paper. The authors mentions specifying ethnicity and age to be specified as root nodes, no specifics on how to do this using pycausal and tetrad was provided however.

### 4.2 Results beyond original paper

The original paper did not include source code to test a different model to DECAF. Therefore, using a causal GAN architecture[7], we generate new data as specified by the causal graph by generating the data conditioned on the attributes specified by said graph. This is done by first calculating the data in the root nodes, and then procedurally generating the nodes that are connected to the previously generated nodes. (Thus taking into account the different privacy measures.) We then calculate the metrics specified in table 2 in the original paper to compare this method to DECAF.

The built-from-scratch DECAF model yielded results that were very similar to the results obtained by DE-CAF model of the original paper. Looking at the table, we can observe a relatively small deviation between the DECAF and imitation FTU and DP metrics.

## 5 Discussion

Our experiments mainly have been conducted to test 1. Most effort we're made in the direction of generating synthetic data corresponding to the underlying distribution. Claim 2 and claim 3 could not be tested yet since claim 1 could not be satisfied.

### 5.1 What was easy

The paper is well written and describes in a clear manner how a causally informed GAN can be used to remove biases and generate fair data. It also explains properly how to satisfy the several fairness conditions for the Adult dataset DAG structure. Thus, this made it easy to get this idea across towards our own DECAF implementation.

Furthermore, the paper provides clear instructions on how to compute the fairness scores, which made it easy to implement.

Additionally, the authors specify that an mlp was used for generating baseline scores. In the actual code however the authors use an xgboost classifier. It was unchallenging to modify the code of the original authors from using an xgboost baseline classifier to a multilayer perceptron.

### 5.2 What was difficult

The main difficulty in replicating the result was in making the model generate synthetic data correctly. Overall the code was difficult to read since there were very little comments provided. The model outputted only single class labels making it impossible for the roc auc score to be computed. Also the authors did not mention that a dag seed and a bias dict needed to be provided to the model so we had to find this out ourselves.

The authors mention that fast greedy equivalence search has been used to discover the variable dependencies in the credit approval dataset. Replicating this result led to difficulties due to insufficient information for how this process was actually done. The authors do not specify how exactly they made this to work because no specification of how the root nodes were provided to the tetrad search algorithm was provided.

Furthermore no code was provided for obtaining the results from the other GAN based models that the original paper mentions the score of. These include the FairGAN score and the Wasserstein GAN with gradient penalty score. These models needed to be replicated from scratch without knowing the hyperparameters that the authors provided.

### 5.3 Communication with original authors

Together with other groups working on the original paper a discussion group was held which was helpful in looking into the other possible approaches for obtaining the results mentioned in the paper. These included looking at the precision and recall scores but leaving the roc auc score gone. Furthermore no contact with the authors was sought.

---

[7]https://arxiv.org/pdf/1907.00503.pdf

# 6 Conclusion

In this paper, we have reproduced the paper *DECAF: Generating Fair Synthetic Data Using Causally-Aware Generative Networks* by van Breugel et al. The scope of this reproduction lies with the results specified in the original paper:

1. We reproduced DECAF, a causal GAN-based model for generating synthetic data.
2. We created a DECAF imitation from scratch, using CTGAN.
3. We tested benchmarks specified by the paper.

First, the reproduction of the main DECAF model was difficult, as the provided code did not run straight away.

Secondly, the DECAF imitation was built using CTGAN, a tabular data generator. This was possible as we could manually specify the causal graph, keeping in mind the privacy definition, and then generate each attribute by training a separate GAN for each node, conditioned on the parent nodes.

Third, the original paper did not include any code to calculate the benchmarks, which meant those had to be created from scratch. The authors later provided the groups working on their paper with more code, which meant we now had an implementation of FairGAN, which was one of the benchmarks. These results have deviated from the original paper quite much.

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
