# OpenReview forum: "[Re] DECAF: Generating Fair Synthetic Data Using Causally-Aware Generative Networks"
_ML_Reproducibility_Challenge/2021/Fall — Reject_

### Official Review · Reviewer_848o · 2022-02-20
**Reproducibility checking of DECAF**

**Rating:** 9
**Confidence:** 5

**Review:**

The authors reproduced DECAF and created a DECAF imitation from scratch, using CTGAN. The authors also tested FAIRGAN in their study. The authors shared relevant links and instructions. So, I believe the work is reproducible. The work is highly significant. The authors described the issues and corresponding solutions properly.

---

### Official Review · Reviewer_Lqc8 · 2022-03-01
**The paper is ok but some statements should be polished**

**Rating:** 4
**Confidence:** 4

**Review:**

Thank you for your great paper!

Summary: The authors tried to reproduce claim 1 of a causal GAN-based model for generating synthetic data in the original DECAF paper and they found that the DECAF evaluations are not reproducible in the same exact manner as the original paper performed. Afterward, they performed their own built-from-scratch DECAF model with CTGAN and found the reproduced paper were similar to the original paper.

Pros:
- The authors calculated the statistics by averaging over ten runs, which makes the results more stable and confident
- The authors managed to find clues about the necessity of a dag seed and a bias dict from the rebuttal of the original paper
- The authors were able to define the main 3 claims of the original paper

Cons:
- Some statements of the paper should be polished

For example,
Section 3.1.2: In order to create some contrast for DECAF -> to compare with DECAF

Section 3.2: 14 different labels which are both continuous and discrete -> not clear

Section 3.4: Both libraries use different versions of tensorboard which leads to a dependecy conflict issue -> should be dependency, and could the conflict be solved by different conda environments?

- The structure of the paper should be reconsidered as it is not concise and precise for readers to understand

From the statement of 4.1.1, it is ambiguous whether Table 1 is the replicated results or the original results. (I think it is the replicated results)

Besides, it is better if the authors could stress their contributions from the beginning of the paper.

---

### Meta-Review · Program_Chairs · 2022-04-07

**Recommendation:** Reject
**Confidence:** 4

**Metareview:**

While the reproduction effort of the DECAF and FairGAN papers is interesting and brings some insights, there are many missing element sto the paper, from the incomplete bibliography to a lack of clarity regarding implementation parameters and results.

---

### Decision · Program_Chairs · 2022-04-09

Reject